# Colistin Selection of the *Mcr-1* Gene in Broiler Chicken Intestinal Microbiota

**DOI:** 10.3390/antibiotics10060677

**Published:** 2021-06-05

**Authors:** Pedro Miguela-Villoldo, Miguel A. Moreno, Agustín Rebollada-Merino, Antonio Rodríguez-Bertos, Marta Hernández, David Rodríguez-Lázaro, Alejandro Gallardo, Alberto Quesada, Joaquín Goyache, Lucas Domínguez, María Ugarte-Ruiz

**Affiliations:** 1VISAVET Health Surveillance Centre, Universidad Complutense de Madrid, Avenida Puerta de Hierro, s/n, 28040 Madrid, Spain; pedromig@ucm.es (P.M.-V.); mamoreno@ucm.es (M.A.M.); agusrebo@ucm.es (A.R.-M.); arbertos@ucm.es (A.R.-B.); jgoyache@ucm.es (J.G.); lucasdo@visavet.ucm.es (L.D.); 2Departamento de Sanidad Animal, Facultad de Veterinaria, Universidad Complutense de Madrid, Avenida Puerta de Hierro, s/n, 28040 Madrid, Spain; 3Departamento de Medicina y Cirugía Animal, Facultad de Veterinaria, Universidad Complutense de Madrid, Avenida Puerta de Hierro, s/n, 28040 Madrid, Spain; 4Laboratorio de Biología Molecular y Microbiología, Instituto Tecnológico Agrario de Castilla y León, Ctra. Burgos Km. 119, 47071 Valladolid, Spain; ita-HerPerMa@itacyl.es; 5Área de Microbiología, Departamento de Biotecnología y Ciencia de los Alimentos, Facultad de Ciencias, Universidad de Burgos, Plaza Misael Bañuelos s/n, 09001 Burgos, Spain; drlazaro@ubu.es; 6Departamento de Bioquímica, Biología Molecular y Genética, Facultad de Veterinaria, Universidad de Extremadura, Avda. de la Universidad s/n, 10003 Caceres, Spain; alexgsoler@unex.es (A.G.); aquesada@unex.es (A.Q.); 7INBIO G+C, Universidad de Extremadura, 0004 Caceres, Spain

**Keywords:** colistin, *mcr-1*, *Salmonella*, in vivo assay, real-time PCR quantification, microbiota, chicks

## Abstract

Colistin has a long story of safe use in animals for the treatment and prevention of certain bacterial diseases. Nevertheless, the first description of the *mcr-1* gene showed that colistin resistance can spread by horizontal gene transfer and changed the landscape. This study aimed to assess the effect of colistin administration on the dispersion of resistance in the microbiota of day-old broiler chicks and how the presence of *mcr-1* genes influences the spread of colistin resistance determinants. In this study, 100 one-day-old chicks were divided into four groups of 25 animals (G1, G2, G3, and G4). Animals from G3/G4 were challenged with *mcr-1*-carrying *Salmonella* (day 7), while colistin (600 mg/L) was administered daily to G2/G4 animals through drinking water (from day 8 to day 15). Two quantitative PCR assays were performed to compare the amount of *Salmonella* and *mcr-1* that were present in the caecal samples. We observed that levels of *mcr-1* were higher in G3/G4 animals, especially G4, due to the spread of *mcr-1*-carrying *Salmonella*. On day 21, *Salmonella* levels decreased in G4, reaching similar values as those for G3, but *mcr-1* levels remained significantly higher, suggesting that colistin may accelerate the spreading process when *mcr-1*-carrying bacteria reach the gut.

## 1. Introduction

The gut microbiome constitutes a rich and diverse microbial ecosystem that influences host nutrition and physiology, developing important functions that are beneficial for host health [1]. The microbiota of new-born chicks, first colonised by bacteria of the facultative anaerobic *Proteobacteria* phylum, progressively changes during the first 19 days of life [2]. Commensal *Enterobacteriaceae* is a major constituent of *Proteobacteria* and plays an important role in the protection of the gut against colonisation by pathogens, such as *Salmonella enterica* (especially serovars Typhimurium and Enteritidis), obligate anaerobic spore-forming bacteria, such as *Clostridia* species, and facultative anaerobic bacteria from the *Streptococcus* and *Staphylococcus* genera [3]. However, *Proteobacteria*, especially *Enterobacteriaceae*, play a critical role in the emergence and maintenance of antimicrobial resistance, as the genes responsible are more prone to be found in this phylum (and also in *Firmicutes*) than in other phyla of the gut microbiome [4]. For example, plasmid-encoded resistance genes can move between gut bacteria via horizontal gene transfer. Thus, commensal *Enterobacteriaceae* become an excellent reservoir for antimicrobial resistance genes, making them accessible to other pathogenic or commensal bacteria [4,5].

Antimicrobial resistance is a public health concern worldwide and has become an increasing threat to human and animal health over the last decade, limiting the therapeutic alternatives available to treat infections caused by multidrug-resistant (MDR) pathogens. The increase of human infections due to multidrug-resistant Gram-negative bacteria, especially those producing extended-spectrum beta-lactamases and carbapenemases, has forced the reintroduction of colistin to treat infections caused by these microorganisms in human medicine, as it is often the only effective antimicrobial against them [6,7,8]. Consequently, colistin has been classified among the antibiotic groups with activity against “Critical Priority” or “High Priority” pathogens identified by the World Health Organisation (WHO) [9] and is considered to be the last line to treat these infections.

The prevalence of colistin resistance in *Enterobacteriaceae* from food animals has been increasing annually and is thought to be related to the use of colistin in veterinary medicine on a global scale [10,11]. Colistin has been used for decades in animals to treat and prevent infectious diseases [6,12,13], usually administered orally in feed or drinking water [14,15]. The Committee for Medicinal Products for Veterinary Use (CVMP) of the European Medicines Agency (EMA) has restricted the administration of colistin sulphate to poultry to 3 to 7 days to treat gastrointestinal infections [16].

Until 2015, all described colistin resistance mechanisms were related to chromosomal point mutations, mainly involving the two-component regulatory system (*pmrAB/phoPQ*) and its negative regulator *mgrB*. However, a new plasmid-mediated gene, called *mcr-1*, was first described in that year, which meant that horizontal transfer was possible [8,17]. Thus, due to the increase of colistin resistance and the emergence of the *mcr-1* gene, the EMA recommended minimising the use of colistin in animals in the European Union (EU) in 2016, aiming to limit its use to 5 mg/population correction unit (PCU) at the national level, with a desirable level of 1 mg/PCU [15]. Since then, colistin resistance levels have appeared to drop with the decrease in its use. We also observed this trend in a recent study carried out in Spain [18].

There have been no studies of the effects of colistin administration on the intestinal microbiota of the neonatal gut and its ability to select colistin-resistant bacteria. Thus, we aimed to assess the effect of colistin administration on the dispersion of resistance in the microbiota of day-old broiler chicks and how the presence of *mcr-1* genes influences the spread of colistin resistance determinants using a monophasic *Salmonella* Typhimurium strain carrying a *mcr-1* gene.

## 2. Results

### 2.1. Clinical Signs and Pathological Findings

Four animals (two animals from G1, one from G3, and one from G4) died during the first week during the adjustment period. Post-mortem examination revealed fibrinous pericarditis and perihepatitis (septicaemic colibacillosis) as the cause of death. We collected samples from these animals, as their deaths occurred the day before the first sampling. Thus, none of the samples were excluded from the experiment. There were no significant changes found in any animal included in the study during the post-mortem examinations throughout the experiment.

### 2.2. Salmonella Counts in the Chick Samples

The amount of *Salmonella* in the chick caecal samples was estimated by plate count. Only animals from the *Salmonella*-challenged groups (G3 and G4) showed growth of *Salmonella* on SMID2 plates (Figure 1). The mean values were non-statistically different (*p* < 0.05): from <5.4 × 10^1^ to 1.2 × 10^5^ CFU/mg (mean of 2.0 × 10^4^ CFU/mg) for G3 (without colistin) and from <5.4 × 10^1^ to 3.6 × 10^5^ CFU/mg (mean of 2.3 × 10^4^ CFU/mg) for G4 (colistin treated). Chicks from G4 had higher *Salmonella* counts from D9 to D14, at which time G4 reached the highest value. Finally, after the withdrawal of colistin (D14), both G3 and G4 showed similar CFU counts on D21 (Figure 1). All recovered *Salmonella* isolates were *mcr-1*-positive by conventional PCR.

### 2.3. Quantitative qPCR of the mcr-1 Gene

In total, 87 of 100 caecal samples were analysed on D7 (7, 20, 20, 20, and 20 samples from the control group, G1, G2, G3, and G4, respectively). Overall, 13 samples were excluded because it was impossible to obtain enough samples (at least 100 mg) for DNA extraction.

Quantitative PCR data for the *mcr-1* gene of the 7 animals from the CTRL group and 21 animals from G1 (neither receiving colistin nor challenged with *Salmonella*) showed high inter-individual variability. Two animals from the CTRL group showed the barely detectable presence of the *mcr-1* gene, which remained below the quantification limit, and four animals from G1 showed quantifiable levels ranging from 6.7 × 10^3^ to 1.6 × 10^6^ *mcr-1* gene copies per mg caecal content, establishing a slight background. The remaining 17 chicks showed no qPCR signals or showed *mcr-1* values below the quantitation limit. The values for the *mcr-1* levels from the groups not challenged with *Salmonella* (G1 and G2) remained similar throughout most of the study, reaching their peaks on D11 (3.2 × 10^5^ and 2.6 × 10^6^ gene copies per mg caecal content, respectively) (Table 1). Concerning the *Salmonella*-challenged groups (G3 and G4), chicks from G3 (colistin untreated) showed an increase from less than 1.0 × 10^2^ to 1.9 × 10^5^ *mcr-1* copies/mg from D9 to D21. The number of *mcr-1* copies/mg for chicks from G4 (colistin treated) increased from 1.5 × 10^3^ to 2.1 × 10^5^ for the same period. The greatest differences between the four groups were observed on D11 and D14, coinciding with the end of colistin administration, with G4 showing significantly higher *mcr-1* values, as the median of these values was outside the interquartile range of the other groups (Figure 2). In addition, chickens from G1 and G2 showed a non-significant increase in the median from D9 to D21 (*p* = 0.23), whereas *mcr-1* levels increased throughout the study in the challenged groups, G3 and G4.

### 2.4. Comparison of Salmonella Spp. and mcr-1 Quantitative qPCR Data

Quantitative PCR data for *Salmonella* quantification for the groups, including chicks inoculated with *mcr-1*-carrying *Salmonella*, showed median values for G3 (without colistin) of <3.3 × 10^2^, 5.1 × 10^3^, 1.5 ×1 0^3^, 1.3 × 10^4^, and 1.1 × 10^4^ copies per mg for days 7, 9, 11, 14, and 21, respectively. For G4 (colistin administered), the corresponding median values were <3.3×10^2^, 1.7 × 10^3^, 1.7 × 10^6^, 4.7 × 10^4^, and 1.6 × 10^4^ copies/mg (Figure 3). Quantification of *mcr-1* and *Salmonella* showed similar values on D9 (*p* < 0.05, Table 2), indicating that most copies of the *mcr-1* genes in the samples came from the inoculated *S*. Typhimurium strain. The levels of *mcr-1* and *Salmonella* increased thereafter in G4, during the colistin administration period, whereas the levels remained approximately the same in G3. After the withdrawal of colistin, *Salmonella* levels in G4 decreased, reaching G3-equivalent values on D21. However, *mcr-1* levels of both groups remained significantly higher (Table 2), suggesting that *mcr-1* may have been transferred to other intestinal bacterial species (Figure 3).

## 3. Discussion

Food-producing animals have been highlighted as potential reservoirs for the dissemination of colistin-resistance determinants, especially since late 2015, when Liu et al. identified the plasmid-mediated colistin resistance gene, *mcr-1*, in China [17]. Horizontal gene transfer processes have been proposed to play a critical role in the spreading of the *mcr-1* gene [19,20]. In the present study, colistin administration had a significant effect on the spread of *mcr*-mediated colistin resistance in the chicks’ microbiota. Its effect was greater when *mcr-1*-carrying bacteria were introduced into the gut environment in the presence of colistin, in which case *mcr-1* colistin resistance appeared to become widespread. We used a *Salmonella enterica* serovar Typhimurium strain because neonatal chicks are highly susceptible to colonisation by *Salmonella* serovars [21] and it is easily identified in chicks because it is not a component of their early microbiota. We studied the effect of colistin by administering it for seven days, as recommended by the EMA for treatment of enteric infections caused by susceptible non-invasive *E. coli* [15], and focused on the *mcr-1* gene because it is the mobile gene most frequently related to colistin resistance worldwide [17].

This scenario has been described in other studies, in which a link was proposed between colistin use and the spread of the *mcr-1* gene [12,18,22]. Comparison of the two groups to which colistin was administered (G2 and G4) showed colistin treatment to significantly increase *mcr-1* levels in G4 (in which the chicks were challenged with *Salmonella*) but not G2 (in which the chicks were not challenged with *Salmonella*).

This study shows the importance of the presence of an intestinal bacterial population that carries the *mcr-1* gene for colistin resistance for the dispersion associated with its use. This is clear from the data of the *mcr-1* challenged chicks which were not administered colistin (G3). Chicks from non-challenged groups (G1 and G2) showed a baseline presence of *mcr-1* from the first day of sampling, which may have been due to the pre-existence of these genes in certain bacterial groups different from *Salmonella*. However, chicks from these groups (G1 and G2) showed parallel progression, with no significant differences in their *mcr-1* outcomes. Chicks from G3, despite not having been given colistin, showed an increase in *mcr-1* levels, which became significantly different from those of the animals from G1 and G2 by D14 and D21. Similarly, in assessing the effect of colistin administration, chicks challenged with *mcr-1*-carrying *Salmonella* (G4) showed significantly more (*p*-value < 0.01) *mcr-1* copies/mg relative to the non-challenged group (G2) (Figure 2), demonstrating that the presence of gene-carrying *Salmonella* in the gut is more decisive in gene dispersal than the effect of the antibiotic, as has been seen in other studies [23].

*Salmonella* counts in chicks from group G4 were higher than those in animals from group G3, as observed for the *Salmonella* qPCR results. Harbouring *mcr* genes usually suppose a negative fitness cost to the carrier bacteria [24] and maybe this fact was responsible for lower levels of *Salmonella* detected in G3 than those from G4, since *Salmonella* in G4 had a selective advantage because of colistin treatment. Thus, data from group G4 support the initial hypothesis about the effect of colistin use on the spread of colistin resistance caused by *mcr* family genes. These data are consistent with the high dispersion capacity of the *mcr-1* gene, especially in the presence of colistin, as previously described [12,13].

Therefore, the *mcr-1* levels of group G4 (colistin administered–*Salmonella* challenged) began to decline after exposure to colistin stopped, but slower than those of *Salmonella*, remaining similar to the values of group G3, which suggests that the reduction corresponded to *mcr-1* harboured by the *Salmonella* and that the remaining *mcr-1* quantified on D21 mainly came from other bacterial species of the microbiota that had been selected by the administration of colistin. Thus, our results highlight the possible horizontal transfer of genes from *Salmonella* to other intestinal bacteria, which may have allowed *mcr-1* levels to remain higher for a longer period in both groups (G3 and G4). Although the clonal spread of *mcr-1*-carrying *Salmonella* is common, there is a close association between certain serotypes of *Salmonella enterica*, as serovar Typhimurium, and *mcr-1* gene and different types of plasmid also play an important role in the conjugation phenomenon [8,25]. Therefore, further studies are needed to determine the type of plasmid where *mcr-1* was located and confirm this hypothesis.

## 4. Methods

### 4.1. Ethical Approval

Experimental procedures were approved by the University Complutense of Madrid Animal Care and Ethics Committee (date of approval: 31/07/2019; registration number: 99/107262.9/19) in compliance with the regulations of the Community of Madrid (PROEX 152/19). Animal experiments took place in the biosafety level 3 (BSL-3) facilities of the VISAVET Health Surveillance Centre and animals were housed according to the European legislation on animal welfare (Directive 2010/63/EU).

### 4.2. Experimental Design

In total, 100 one-day-old broiler chicks (Ross 308) were obtained from a commercial hatchery and housed at the VISAVET Surveillance Centre facilities for the 21 days of the experiment under the same environmental conditions described previously by Herrero-Encinas et al. [26]. On the first day, when the animals arrived, box litter samples were collected for *Salmonella* spp. detection following ISO 6579:2017 standards. All samples were found negative. Chicks were divided into four groups of 25 animals. All chicks were housed in different cages until day (D) 7 but under the same environmental and feeding conditions. Thus, they were considered to be a single group for data analysis (control group: CTRL). From D7, chicks from each cage were exposed to different conditions to form four groups: animals without colistin nor *Salmonella* challenge (G1), colistin administered (G2), *Salmonella* challenged (G3), and colistin administered and *Salmonella* challenged (G4). On D7, chicks from G3 and G4 were challenged with a *Salmonella enterica* subsp. *enterica* serovar Typhimurium (monophasic variant) strain that was positive for *mcr-1* through their drinking water (3.3 × 10^5^ CFU/mL). Starting from D8, colistin (colistin sulphate 1,025,000 IU, Acolan, Spain) was administered for seven days to chickens of groups G2 and G4 through their drinking water (600 mg/L) adjusting it to a concentration of 75,000 IU recommended in poultry by the EMA [15], replacing the water and corresponding dose of colistin every 24 h.

### 4.3. Sampling and Sample Preparation

On days 7, 9, 11, 14, and 21, five randomly selected animals from each group (CTRL and G1 to G4) were sedated intramuscularly with diazepam and euthanised with an overdose of sodium pentobarbital by intraperitoneal injection. Caecum faeces were collected during the autopsy, and an aliquot of 1.5 g was stored at 4 °C for analysis using microbiological methods in the following 24 h. The remaining caecal faeces collected from each animal was preserved at −80 °C for molecular analysis.

#### Salmonella Counting Using Selective Media

A gram of each fresh aliquot was mixed with 2 mL saline (0.85% NaCl). Then, 50 µL was diluted into 9 mL brain–heart infusion broth (BHI) supplemented with a 10 µg colistin disk and incubated at 37 °C for 4 h. Then, six 10-fold serial dilutions were carried out in BHI. ChromID selective medium *Salmonella* Agar (SMID2) (bioMérieux, Marcy-l’Étoile, France) was used for *Salmonella* counting. SMID2 plates were inoculated with 100 µL of −3 and −4 BHI sample dilutions and incubated at 37 °C for 24 h. After incubation, all colonies suspected to be *Salmonella* were counted according to the manufacturer’s specifications. One colony of each SMID2 plate was streaked onto blood agar plates and incubated at 37 °C for 24 h for subsequent species confirmation by mass spectrometry using a Bruker Daltonics UltrafleXtrem MALDI TOF/TOF instrument (Bruker Daltonics, Bremen, Germany). Conventional PCR was performed to confirm the presence of the *mcr-1* gene [27].

### 4.4. Quantitative Assay for Mcr-1

Direct DNA extraction from chick caecal samples was carried out using a commercial kit (FASTI001-1 FavorPrep Stool DNA Isolation Mini Kit, Favorgen-Europe, Vienna, Austria) following the manufacturer’s specifications (elution volume of 200 µL), coupled with a specific SYBRGreen (Thermo Fisher Scientific, Vilnius) real-time PCR assay for quantitative detection of the *mcr-1* gene (qPCR), as described previously by Li J et al. [28] and further validated in our previous work [29]. Samples were considered positive if quantitative values were >1.00 × 10^2^ fg/µL (equivalent to 1.58 × 10^3^ copies/mg caecal content).

### 4.5. Quantitative Real-Time PCR Assay for Salmonella

Quantitative real-time PCR was carried out for the quantitative detection of *Salmonella* spp. in each sample using a commercial “*Salmonella* spp. DNA extraction and real-time PCR detection” kit (Kylt^®^ *Salmonella* spp. (FS), Oldenburg, Germany). Two microliters of each DNA elute were also run in triplicate. A sample was considered to be positive when its cycle threshold (CT) was ≤42.

### 4.6. Statistical Analysis

The data obtained by qPCR were analysed using a t-test for two related samples after normalisation by logarithmic transformation into Log10. A Kruskal–Wallis test was used to analyse differences in SMID2 *Salmonella* counts among experimental groups. A difference was considered significant when the *p*-value was <0.05 for both statistical tests.

### 4.7. Data Visualisation

All figures included in this study were illustrated with R (core team 2019) [30] using the ggplot2 package (H. Wickham, 2016) [31].

## 5. Conclusions

The presence of *mcr-1*-carrying *S*. *enterica* serovar Typhimurium in the gut resulted in the spread of the *mcr-1* gene, probably due to horizontal gene transfer. The administration of colistin accelerated this process by selecting the colistin-resistant bacteria present in the gut microbiota, keeping *mcr-1* levels constant after the withdrawal of colistin.

## Figures and Tables

**Figure 1 antibiotics-10-00677-f001:**
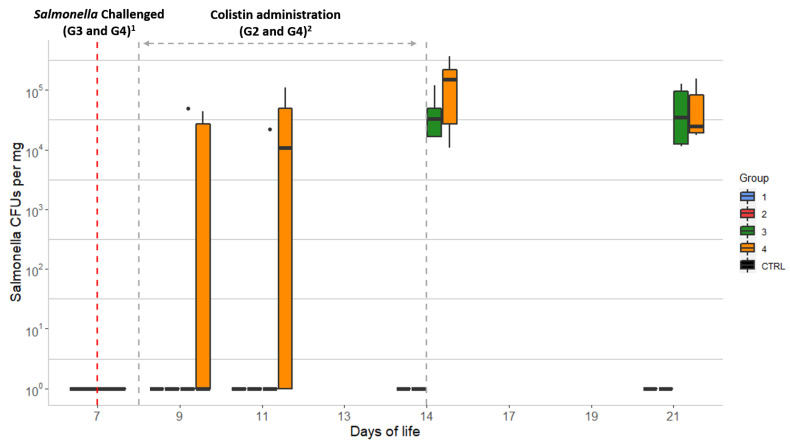
Evolution of *Salmonella* counts over 21 days in chick caecal samples: CTRL: control group; Group 1 (G1): animals without colistin nor *Salmonella* challenge; Group 2 (G2): colistin administered; Group 3 (G3): *Salmonella* challenged; Group 4 (G4): colistin administered and *Salmonella* challenged. Dots represent outliers; ^1^
*Salmonella* challenge started at D7 for groups G3 and G4 (sampling from D7 was performed before the challenge was started); ^2^ Colistin was provided to chicks from groups G2 and G4 from D8 to D14.

**Figure 2 antibiotics-10-00677-f002:**
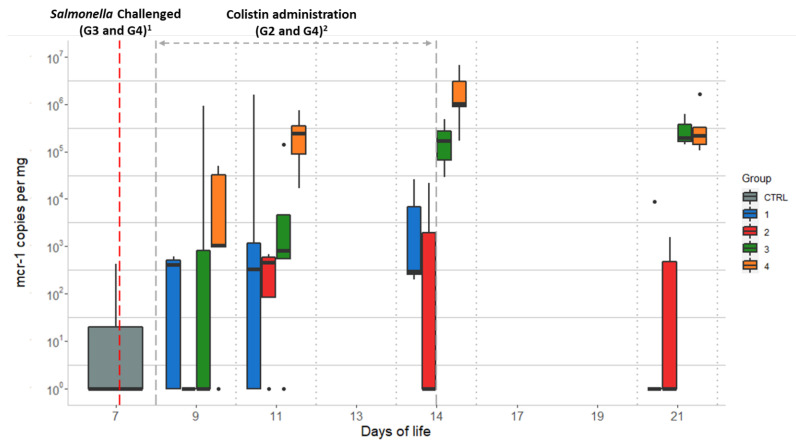
Quantitative qPCR data of *mcr-1* gene copies/mg obtained per day of life of the chicks for each studied group: Group 1 (G1): animals without colistin nor *Salmonella* challenge; Group 2 (G2): colistin administered; Group 3 (G3): *Salmonella* challenged; Group 4 (G4): colistin administered and *Salmonella* challenged. Dots represent outliers; ^1^ *Salmonella* challenge started at D7 for groups G3 and G4 (sampling from D7 was performed before starting the challenge); ^2^ Colistin was provided to chicks from groups G2 and G4 from D8 to D14.

**Figure 3 antibiotics-10-00677-f003:**
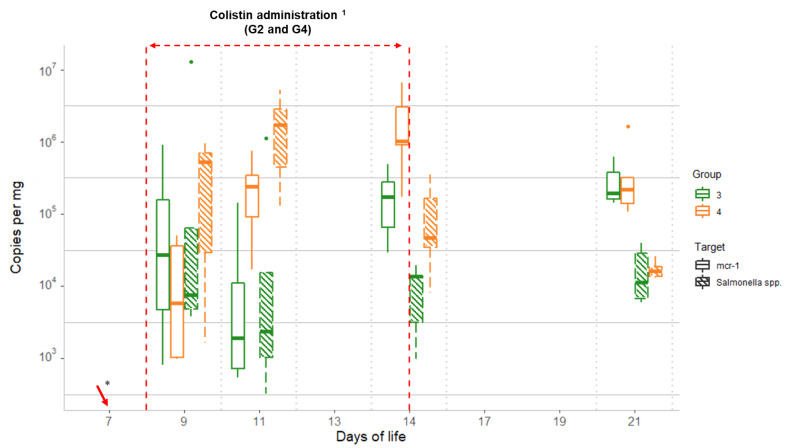
Comparison between qPCR data for *Salmonella* and *mcr-1* obtained throughout the study for the challenged groups (G3 and G4): Group 3 (G3): *Salmonella* challenged; Group 4 (G4): colistin administered and *Salmonella* challenged. Dots represent outliers; ^1^ Colistin was provided to chicks from group G4, from D8 to D14; * *Salmonella* challenge started on D7 for groups G3 and G4 (sampling from D7 was performed before starting the challenge).

**Table 1 antibiotics-10-00677-t001:** Quantitative PCR data for *mcr-1* gene copies/mg per day of life (expressed in Log10).

*mcr-1* Gene Copies/mg (Expressed in Log10)												
	SC (D7)	CA (D8)	Day 7 ^1^		Day 9		Day 11		Day 14		Day 21	
			$\bar{x}$	Me	$\bar{x}$	Me	$\bar{x}$	Me	$\bar{x}$	Me	$\bar{x}$	Me
CTRL ^2^	No	No	2.07	<2.0	NA	NA	NA	NA	NA	NA	NA	NA
G1	No	No	NA	NA	2.49	2.60	5.51	2.52	3.83	2.47	3.24	<2.0
G2	No	Yes	NA	NA	<2.0	<2.0	6.43	2.75	3.67	<2.0	2.60	<2.0
G3	Yes	No	NA	NA	5.26	<2.0	4.47	2.90	5.32	5.24	5.47	5.29
G4	Yes	Yes	NA	NA	4.23	3.02	5.46	5.38	6.37	6.01	5.68	5.34

SC (D7): *Salmonella* challenge (at day 7), CA (D8): Colistin administration (colistin treatment started at day 8), $\bar{x}$: Mean, Me: median; NA: data were not analysed due to the absence of animals; CTRL: control group; G1: animals without colistin nor *Salmonella* challenge; G2: colistin administered; G3: *Salmonella* challenged; G4: colistin administered and *Salmonella* challenged; ^1^ the CTRL samples were composed of a different number of animals due to the inability to obtain sufficient samples from 13 chicks. Five animals per group were sampled for the other days; ^2^ CTRL data only available for D7. After D7, chicks were considered from different groups (G1 to G4), depending on the experimental conditions.

**Table 2 antibiotics-10-00677-t002:** T-test *p*-values obtained from a comparative analysis of *mcr-1* and *Salmonella* qPCR quantitative data per sampling day.

		*p*-Values				
qPCR Data	Group	D7	D9	D11	D14	D21
*mcr-1*	G3 vs. G4	ND	0.41	0.03 *	0.02 *	0.86
*Salmonella*	G3 vs. G4	ND	0.69	0.02 *	0.03 *	0.69
*mcr-1* vs. *Salmonella*	G3	ND	0.06	0.44 *	<0.01 *	<0.01 *
*mcr-1* vs. *Salmonella*	G4	ND	0.94	<0.01 *	<0.01 *	<0.01 *

* Statistically significant differences (*p* < 0.05); G3: *Salmonella* challenged; G4: colistin administered and *Salmonella* challenged.; Sampling days: D7: day 7; D9: day 9; D11: day 11; D14: day 14; D21: day 21.; ND: The *t*-test was not applied for data from D7 due to the different number of samples available from each group.

## Data Availability

The data presented in this study are available on request from the corresponding author.

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
