# Peer review of "Colistin Selection of the Mcr-1 Gene in Broiler Chicken Intestinal Microbiota"

_antibiotics, 2021, doi:10.3390/antibiotics10060677_

Round 1

Reviewer 1 Report

In this manuscript, an attempt was made to select bacteria that present the mcr-1 gene by aggregating bacteria carrying the mcr-1 gene and colistin through water in broiler chicken.

The following comments are made.

  1. English revision by a native English speaker.
  2. Lines 29-36 not clear what you mean.
  3. Line 47. Very important: put the scientific names of the bacteria correctly. They are in italics and the species in lowercase. Correct in all text.
  4. Line 73. Genes are written in italics. You should also explain who the mcr-1 gene is and its importance.
  5. Line 80. Because one-day-old neonates, is it normal to administer the antibiotic to this type of animal?
  6. Lines 99-100. Write the scientific notation well. Correct in all text. According to Figure 1 the values ​​are less than 105 CFU / mg. How can you be sure that your values ​​are true if your detection method involves the growth of colistin resistant bacteria?
  7. Figure 1. What do the asterisks mean?
  8. Table 1. According to their experimental design, the Control Group and Group 1 do not have any difference, they are not treated with anything, how do you explain the differences obtained? Why does Group 1 have resistance genes if it is not treated?
  9. Line 130-134. There are no significant differences between Group 3 and Group 4, how do you explain this?
  10. Line 168. How do you explain that the amount of Salmonella decreases by removing the colistin? The decline in Salmonella copies begins to decline before the end of colistin treatment. The data is not clear.
  11. Line 170. How can you assure this statement, which experiment supported this? What other species are there?
  12. Figure 3. What is the difference between Figures a) and b)? You can leave only Figure b).
  13. Line 215-216. This is logical since the Salmonella strain used had the mcr-1 gene.
  14. Line 225. There is an increase in mcr-1 genes, because you infected with bacteria that carry it and, as they replicate, the number of genes increases.
  15. Lines 228-230. Where do these results show?
  16. Line 250-251. You cannot give this conclusion until it is proven.
  17. Line 267. It was 100 chickens of one day old or 100 day old chickens, it is not clear.
  18. Line 280. Water with 3.3X105 CFU / mL of S. typhimurium, correct units and name of bacteria. As you controlled the amount of bacteria that the chickens ingested, it was not an equal dose in all the chickens since it was in the water and each chicken consumes a different amount of water.
  19. Line 282. You also did not control the dose of colistin since you do not know how much water the chickens drank.
  20. Line 290. Correct 4oC.
  21. Line 296. By incubating the sample with colistin you are selecting for the growth of resistant bacteria. So your data is not real.
  22. Line 299. Are they 10-3 and 10-4 dilutions? If so correct it.
  23. Line 305. Reference 23 is a multiplex and you only detected one gene, you could clearly specify the methodology used including primers and conditions.
  24. Line 314. You could write the scientific notation appropriately throughout the text.
  25. Many incomplete References, for example: 2 which Journal ?; 4, pages ?; 13 what Journal ?; 23 what pages ?, etc. Check all references.

Reviewer 2 Report

Review “Colistin selection of the mcr-1 gene in broiler chicken intestinal microbiota”

The authors present data on the supplementation of colistin and mcr-1 positive Salmonella to a neonatal chicken model, and their data suggests that colistin treatment in combination with the presence of mcr-1 carrying Salmonella, leads to overgrowth of the resistant Salmonella in the colistin treated animals.

The paper is well written, and data clearly represented.

A few minor comments are:

  • Lines 79-80. Recently, a paper was published on colistin treatment in weaner pigs. It would be advisable to discuss. (Ahmed, S., Hansen, C., Dahlkilde, A.L., Herrero-Fresno, A., Pedersen, K.S., Nielsen, J.P. and Olsen, J.E., 2021. The Effect of Colistin Treatment on the Selection of Colistin-Resistant Escherichia coli in Weaner Pigs. Antibiotics, 10(4), p.465.).
  • Lines 239-247. It might be worthwhile to discuss mcr-1 carrying plasmid stability. In Klebsiella there is a fitness cost for mcr-1 carrying plasmids (Nang, S.C., Morris, F.C., McDonald, M.J., Han, M.L., Wang, J., Strugnell, R.A., Velkov, T. and Li, J., 2018. Fitness cost of mcr-1-mediated polymyxin resistance in Klebsiella pneumoniae. Journal of Antimicrobial Chemotherapy, 73(6), pp.1604-1610.). Does this also apply to Salmonella?
  • If mcr-1 is horizontally transferred to other bacteria, it is recommendable to discuss Cui, M., Zhang, J., Zhang, C., Li, R., Chan, E.W.C., Wu, C., Wu, C. and Chen, S., 2017. Distinct mechanisms of acquisition of mcr-1–bearing plasmid by Salmonella strains recovered from animals and food samples. Scientific reports, 7(1), pp.1-9.
  • Might be worthwhile to cite: Lima, T., Domingues, S. and Da Silva, G.J., 2019. Plasmid-mediated colistin resistance in Salmonella enterica: a review. Microorganisms, 7(2), p.55.

Round 2

Reviewer 1 Report

  1. Line 47, 207, 253, 259: Again you are misspelling the scientific names of bacteria
  2. The final page is still missing in several references such as: 4, 8, 10,17,23,25,27, and 28.

Author Response

  • Line 47, 207, 253, 259: Again you are misspelling the scientific names of bacteria

Answer: we have modified the scientific name of Salmonella adding the species (enterica) on lines 207, 253 and 259.

Regarding the spelling for serovars, we are using the current rule “The name of the serovar is given in nonitalicised Roman alphabet letters with the first letter capitalized”, according to Ryan et al. (2017).

    • Ryan MP, O'Dwyer J, Adley CC. Evaluation of the Complex Nomenclature of the Clinically and Veterinary Significant Pathogen Salmonella. Biomed Res Int. 2017;2017:3782182

This is also used, for instance, in the Kauffmann-White scheme [Grimont PAD et al. (2008) and Guibourdenche M, et al. (2010)].

    • Patrick AD. Grimont F-XW. Antigentic Formulae of the Salmonella Serovars. 9 ed: WHO Collaborating Centre for Reference and Research on Salmonella; 2008.
    • Guibourdenche M, Roggentin P, Mikoleit M, Fields PI, Bockemühl J, Grimont PAD, et al. Supplement 2003–2007 (No. 47) to the White-Kauffmann-Le Minor scheme. Research in Microbiology. 2010;161(1):26-9.
  • The final page is still missing in several references such as: 4, 8, 10,17,23,25,27, and 28.

Answer: all references final pages were reviewed and corrected in the text.